# Experimental Modeling of Pressure in the Hydrostatic Formation of a Cylindrical Cup with Different Materials

**Trung-Kien Le** [1] [iD], **Thi-Thu Nguyen** [1,*] [iD] and **Ngoc-Tam Bui** [1,2,*] [iD]

1  School of Mechanical Engineering, Hanoi University of Science and Technology, Hanoi 10000, Vietnam; kien.letrung@hust.edu.vn
2  College of Systems Engineering and Science, Shibaura Institute of Technology, Tokyo 135-8548, Japan
*  Correspondence: thu.nguyenthi@hust.edu.vn (T.-T.N.); tambn@shibaura-it.ac.jp (N.-T.B.)

**Abstract:** Forming complex sheet products using hydrostatic forming technology is currently a focus of the majority of forming processes. However, in order to increase stability during the forming process, it is necessary to identify and analyze the dependency of the forming pressure and the quality of a product on input parameters. For the purpose of modeling the forming pressure, this paper presents empirical research on the product of a cylindrical cup made of various materials, including carbon steel (DC04), copper (CDA260), and stainless steel (SUS 304) with different thicknesses (0.8 mm, 1.0 mm, and 1.2 mm), under a defined range of binder pressures. The regression method is selected to formulate an equation that shows the relationship between the input parameters, including the materials (ultimate strength and yield stress), workpiece thickness, binder pressure and the output parameter, and the formation of fluid pressure. The mathematical equation allows us to determine the extent of the effect of each input on the forming pressure. The experimental results can be used for the easier planning and forecasting of the process and product quality in hydrostatic forming.

**Keywords:** hydroforming; forming pressure; sheet metal forming; regression; experimental modeling



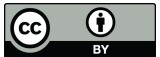

## 1. Introduction

The strong development of automotive and machine industries over the past decades has required advanced manufacturing techniques, especially for thin-shell products. Therefore, a hydrostatic forming technology with outstanding characteristics for the production of high-quality sheet details has been researched and applied in industrial manufacturing [1–7]. The process of hydrostatic formation is fast and satisfies the forming requirements of thin-shell parts. The working principle of this technology is shown in Figure 1. Moreover, this technology can be suitable for the shaping of a wide variety of materials, from high deformability materials, such as carbon steel material, to low deformability materials, such as stainless steel or some alloys [1,8–12].

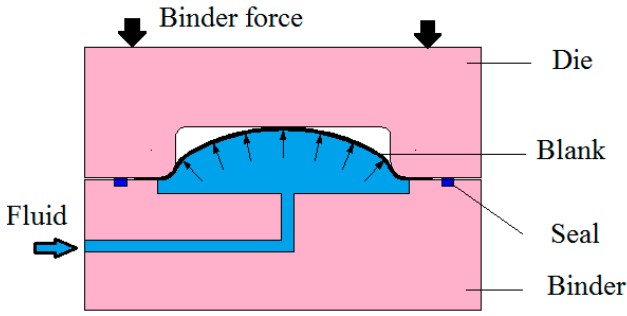

**Figure 1.** Hydrostatic formation process diagram.

There are many impressive studies on forming technology. In this context, the main concerns are technological parameters such as the forming fluid pressure, closing pressure, shape of die, friction, workpiece material, etc. [3,13–19]. Among them, a large amount of research on forming liquid pressure parameters has been published to determine their influence on the forming process and product quality [20–24]. Studies have been conducted with many methods, the most common of which have been simulation or experimentation, or a combination of both. The results of these studies were impressive, informative, and highly applicable. Karabegovic, E. and Poljak, J. [25] worked to obtain a mathematical model of the forming fluid pressure. Their study built a mathematical relationship between the working fluid pressure and the thickness of steel and aluminum in the hydrostatic forming process for pairs of welded sheets. Assempour and Emami [26] also studied the control of forming pressure in hydraulic formation. Modi et al. optimized the forming fluid pressure using the Taguchi method when forming an AA 5182 alloy material to shape squared-cup parts [27]. The influence of technological parameters on product quality was also studied in the work of Fitsum et al. [12]. These authors demonstrated that the formability of high-strength cryo-rolled Al alloy sheets could be enhanced by hydroforming.

Thu N.T. et al. studied the influence of working pressure on the product quality in the hydrostatic formation of DC04 materials [28,29]. A relationship between technological parameters (working pressure and blank holder pressure) and the die geometry has been proposed to optimize the forming process for the cylindrical cup. However, the majority of studies have been conducted with only one type of material, while very few authors have implemented different types of materials. Therefore, the comparison of the effects of forming pressure on products with different materials has been limited.

This paper provides an analysis of the forming process of a typical cylindrical cup in hydrostatic forming with three types of materials—steel (DC04), copper (CDA260), and stainless steel (SUS 304) at different thicknesses (0.8 mm, 1.0 mm, 1.2 mm)—within the binder pressure range of 80~115 bar. For each experimental product with a defined shape and size, the forming pressure is measured. Based on the results of the experiments, the mathematical equation of forming pressure is formulated. The formula allows for the analysis of the influence of each input variable (binder pressure, thickness of material, and ratio of stress (ultimate strength per yield stress)) on forming pressure.

## 2. Research Object and Methodology

The purpose of this research was to determine the relationship between forming pressure and other parameters, including the binder pressure ($Q$), thickness of workpiece ($s_0$), and the ratio of stress ($\sigma$). Thus, a mathematical model of the forming pressure is given in Equation (1).

$$P = f(Q, s_0, \sigma) \tag{1}$$

Binder pressure is one of the key factors in the hydrostatic forming process. It is both effective in preventing coronary instability and also maintaining the forming pressure during the forming process. Thus, it was selected as a factor for research in this paper. Based on the diameter of a product, the binder pressure range was from 80 bar to 115 bar.

The initial blank thicknesses of each material was 0.8 mm, 1.0 mm, and 1.2 mm, with the same diameter of 110 mm, as shown in Figure 2. This is a popular thickness range used in thin-shell forming technology.

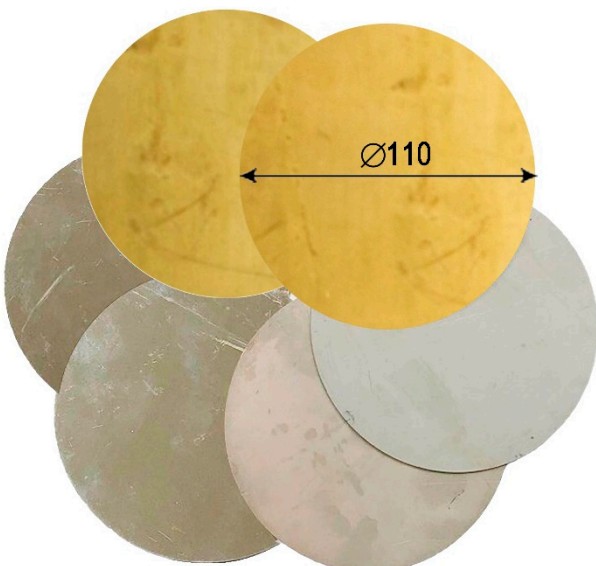

**Figure 2.** Experimental workpieces.

Materials selected for the survey were steel DC04, copper CDA260, and sus 304. The ultimate strength, yield stress, and degree of tensile strain of each material represent that material's ability to deform. Here, to consider the mathematical equation, two parameters—the ultimate strength and yield stress—are used to represent the material under investigation. The ultimate strength and the yield stress of each material were determined and are presented in Table 1.

**Table 1.** Properties of materials.

| Materials | Ultimate Strength $\sigma_m$ (MPa) | Yield Stress $\sigma_f$ (MPa) | Ratio of Stress $\sigma$ |
|---|---|---|---|
| Cu | 420 | 320 | 1.3 |
| DC04 | 415 | 220 | 1.9 |
| SUS 304 | 520 | 205 | 2.5 |

In this study, the ratio of stress of these materials was selected as a representative parameter for each material. This is defined as the ratio between ultimate strength ($\sigma_m$) and yield stress ($\sigma_f$), as shown in Equation (2).

$$\sigma = \frac{\sigma_m}{\sigma_f} \tag{2}$$

The selected product was a low, cylindrical cup. The product was shaped according to the size of the die shown in Figure 3. According to the hydrostatic forming mechanism, the most difficult position to shape was the bottom radius of the die. The bottom radius of the product was highly dependent on the strength of the forming liquid pressure.

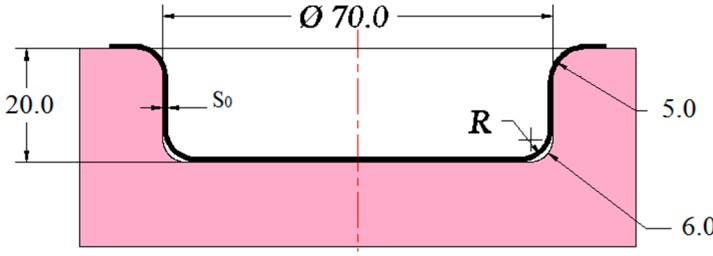

**Figure 3.** Die profile.

The method chosen in this study was empirical planning. With the subjects and the research conditions presented above, an orthogonal second order design for three variables was considered to be most suitable to solve the issue.

## 3. Experiment Process and Results

The experimental diagram is presented in Figure 4 with four main modules:

+ A hydraulic 125-ton press (7), which applies a fixed die (10) and binder (9).
+ A CP700 high pressure pump generating a forming pressure P up to 700 bar—the working fluid is oil with a hydraulic label of AW 46 and density of 0.875 g/cm³ (200 °C).
+ A mold system, including a die (10) and binder (9). Actual pictures of these parts are shown in Figure 5a.
+ A measurement system to measure the forming pressure, binder pressure and product height with stroke sensor (11) and pressure sensors (8, 12) connected to a computer. Actual pictures of these parts are presented in Figure 5b.

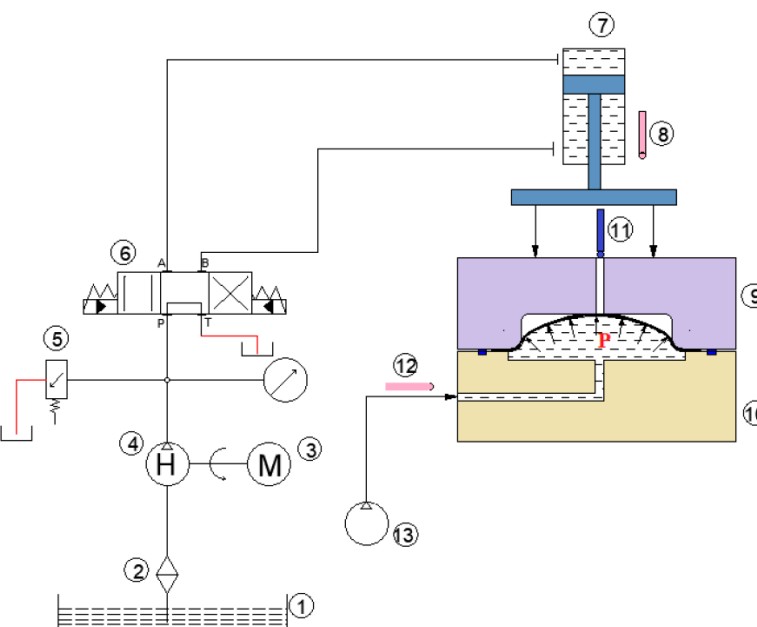

1. Oil tank
2. Oil filter
3. Engine
4. Pump 1
5. Safety valve
6. Distribution valve
7. Hydraulic press
8. Pressure sensor 1
9. Upper die
10. Lower die
11. Stroke sensor
12. Pressure sensor 2
13. CP700 high pressure pump

**Figure 4.** Experiment diagram.

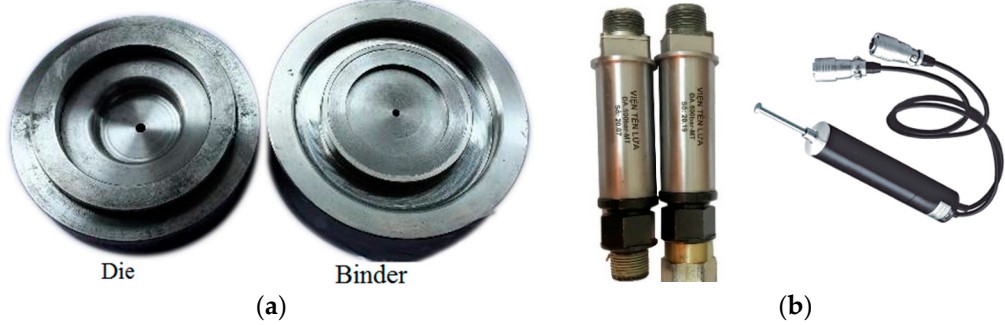

Die        Binder
(**a**)                    (**b**)

**Figure 5.** Components of the experimental system. (**a**) Hydrostatic die and binder (lower die); (**b**) Pressure sensors, stroke sensor.

The experimental system is shown in Figure 6. Under the effect of liquid pressure, the workpiece was deformed and shaped according to the cavity of the die. The parameters of the binder pressure, the forming pressure, and the depth of the product were identified

and displayed on a measuring system connected to a computer during the forming process. The forming ended when the liquid pressure reached the investigated value, the workpiece tore, or the system failed to maintain the fluid pressure.

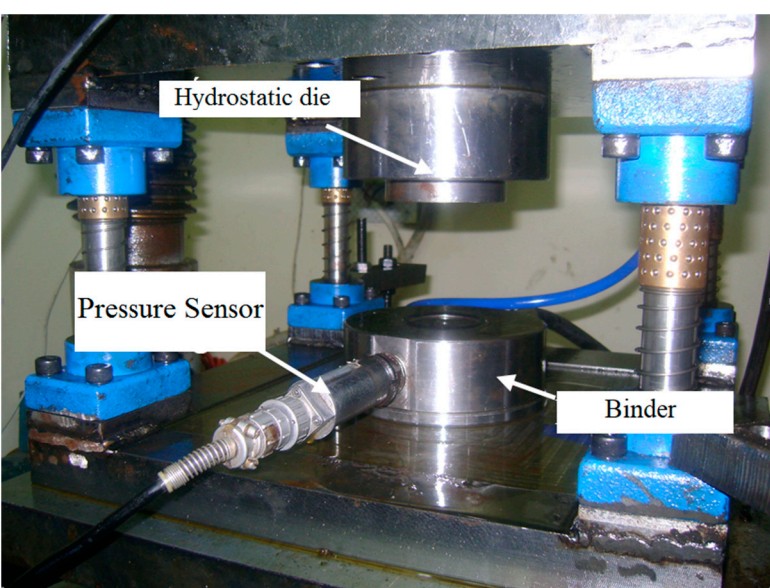

**Figure 6.** Experimental system.

Initial tests showed that the forming pressure had a great influence on the geometry of the product, especially the bottom radius of the product's curvature. The greater the pressure, the closer the radius was brought to the die. Figure 7 shows the effect of the fluid pressure strength on the product forming ability. Therefore, the forming pressure was a factor that needed to be clarified.

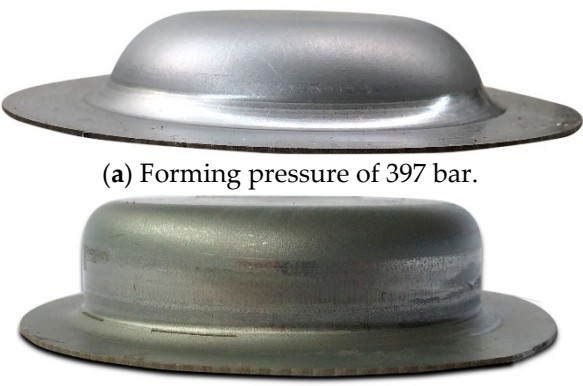

(**a**) Forming pressure of 397 bar.

(**b**) Forming pressure of 566 bar.

**Figure 7.** Products at different forming pressures.

After setting up the experimental diagram as shown above, the experimental process was conducted according to the experimental array of the orthogonal second-order design method [30]. The number of experiments was determined based on formula (3) as follows:

$$N = 2^k + 2k + n_0 = 2^3 + 2.3 + 3 = 17 \tag{3}$$

where N is the total number of experiments, k is the number of variables surveyed, and $n_0$ is the number of repetitions in the central plan.

The experimental array was built according to Table 2. The product was shaped as shown in Figure 8.

**Table 2.** Experimental array.

| | Physical Values | | | Coded Values | | | | | | | | | | |
|---|---|---|---|---|---|---|---|---|---|---|---|---|---|---|
| No | Q | $S_0$ | σ | $x_0$ | $x_1$ | $x_2$ | $x_3$ | $x_{12}$ | $x_{13}$ | $x_{23}$ | $x'_1$ | $x'_2$ | $x'_3$ | Output Pi |
| 1 | 80 | 0.8 | 1.3 | 1 | −1 | −1 | −1 | 1 | 1 | 1 | 0.27 | 0.27 | 0.27 | $Y_1$ |
| 2 | 115 | 0.8 | 1.3 | 1 | 1 | −1 | −1 | −1 | −1 | 1 | 0.27 | 0.27 | 0.27 | $Y_2$ |
| 3 | 80 | 1.2 | 1.3 | 1 | −1 | 1 | −1 | −1 | 1 | −1 | 0.27 | 0.27 | 0.27 | $Y_3$ |
| 4 | 115 | 1.2 | 1.3 | 1 | 1 | 1 | −1 | 1 | −1 | −1 | 0.27 | 0.27 | 0.27 | $Y_4$ |
| 5 | 80 | 0.8 | 2.5 | 1 | −1 | −1 | 1 | 1 | −1 | −1 | 0.27 | 0.27 | 0.27 | $Y_5$ |
| 6 | 115 | 0.8 | 2.5 | 1 | 1 | −1 | 1 | −1 | 1 | −1 | 0.27 | 0.27 | 0.27 | $Y_6$ |
| 7 | 80 | 1.2 | 2.5 | 1 | −1 | 1 | 1 | −1 | −1 | 1 | 0.27 | 0.27 | 0.27 | $Y_7$ |
| 8 | 115 | 1.2 | 2.5 | 1 | 1 | 1 | 1 | 1 | 1 | 1 | 0.27 | 0.27 | 0.27 | $Y_8$ |
| 9 | 97.5 | 1 | 1.9 | 1 | 0 | 0 | 0 | 0 | 0 | 0 | −0.73 | −0.73 | −0.73 | $Y_9$ |
| 10 | 97.5 | 1 | 1.9 | 1 | 0 | 0 | 0 | 0 | 0 | 0 | −0.73 | −0.73 | −0.73 | $Y_{10}$ |
| 11 | 97.5 | 1 | 1.9 | 1 | 0 | 0 | 0 | 0 | 0 | 0 | −0.73 | −0.73 | −0.73 | $Y_{11}$ |
| 12 | 119 | 1 | 1.9 | 1 | 1.215 | 0 | 0 | 0 | 0 | 0 | 0.75 | −0.73 | −0.73 | $Y_{12}$ |
| 13 | 76 | 1 | 1.9 | 1 | −1.22 | 0 | 0 | 0 | 0 | 0 | 0.75 | −0.73 | −0.73 | $Y_{13}$ |
| 14 | 97.5 | 1.24 | 1.9 | 1 | 0 | 1.22 | 0 | 0 | 0 | 0 | −0.73 | 0.75 | −0.73 | $Y_{14}$ |
| 15 | 97.5 | 0.76 | 1.9 | 1 | 0 | −1.2 | 0 | 0 | 0 | 0 | −0.73 | 0.75 | −0.73 | $Y_{15}$ |
| 16 | 97.5 | 1 | 2.6 | 1 | 0 | 0 | 1.215 | 0 | 0 | 0 | −0.73 | −0.73 | 0.75 | $Y_{16}$ |
| 17 | 97.5 | 1 | 1.2 | 1 | 0 | 0 | −1.22 | 0 | 0 | 0 | −0.73 | −0.73 | 0.75 | $Y_{17}$ |

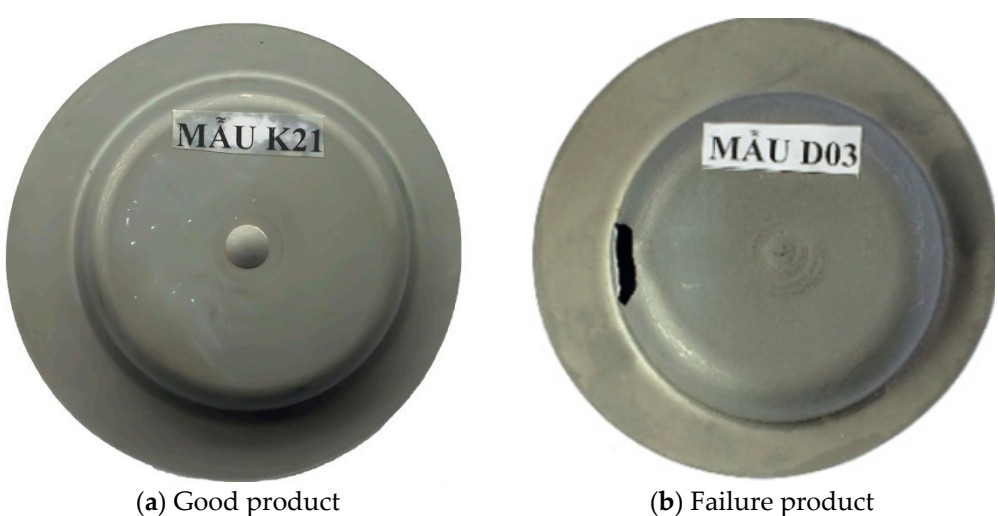

(**a**) Good product          (**b**) Failure product

**Figure 8.** Experimental products.

The maximum forming liquid pressure reached in each experiment—denoted by $P_{max}$—is summarized in Table 3. Pressurization only took place strongly in the final stage, which was the stage of product calibration following the die profile.

**Table 3.** Experimental results.

| $N_0$ | 1 | 2 | 3 | 4 | 5 | 6 | 7 | 8 | 9 | 10 | 11 | 12 | 13 | 14 | 15 | 15 | 17 |
|---|---|---|---|---|---|---|---|---|---|---|---|---|---|---|---|---|---|
| $P_{max}$ (bar) | 397 | 490 | 449 | 620 | 436 | 647 | 429 | 672 | 537 | 546 | 529 | 566 | 397 | 573 | 507 | 548 | 451 |



## 4. Mathematical Modeling of Forming Pressure

Based on the data in Table 3, a mathematical model describing the effect of the input parameters on the forming pressure was built. Equation (1) is rewritten as Equations (4) and (5).

$$P_{max} = f(Q, S_0, \sigma) \tag{4}$$

$$i.e \ P_{max} = Y_i = f(x_1, x_2, x_3) \tag{5}$$

where the physical variable is converted into a coded variable in Equation (6). The values of the coding variables are given in Table 4.

$$x_1 = \frac{Q - 97.5}{17.5} \ ; \ x_2 = \frac{S_0 - 1}{0.2} \ ; \ x_3 = \frac{\sigma - 1.9}{0.6} \tag{6}$$

where $x_1$, $x_2$, and $x_3$ are coded variables of the binder pressure, the thickness of the workpiece, and the ratio of stress, respectively.

**Table 4.** Input parameters used in the research.

| Input Parameters | Coded Values | | |
|---|---|---|---|
| | **−1** | **0** | **+1** |
| $x_1 - Q$ | 80.0 | 97.5 | 115.0 |
| $x_2 - S_0$ | 0.8 | 1.0 | 1.2 |
| $x_3 - \sigma$ | 1.3 | 1.9 | 2.5 |

According to orthogonal second-order design theory [30], the formula of forming pressure is represented by Equation (7):

$$Y = b_0 + \sum_{j=1}^{2} b_j x_j + \sum_{i,j=1; i \neq j}^{2} b_{ij} x_i x_j + \sum_{j=1}^{2} b_{jj} x_j^2 \tag{7}$$

where $b_0$, $b_j$, $b_{ij}$, and $b_{jj}$ are the coefficients of the regression function and are determined by Equations (8)–(11).

$$b_0 = \frac{1}{n} \sum_{u=1}^{n} Y_u \tag{8}$$

$$b_j = \frac{\sum_{u=1}^{n} x_{uj} Y_u}{\sum_{u=1}^{n} x_{uj}^2} \tag{9}$$

$$b_{ij} = \frac{\sum_{u=1}^{n} (x_{ui} x_{uj}) Y_u}{\sum_{u=1}^{n} (x_{ui} x_{uj})^2} \tag{10}$$

$$b_{jj} = \frac{\sum_{u=1}^{n} x'_{ui} Y_u}{\sum_{u=1}^{n} (x'_{ui})^2} \tag{11}$$

The coefficients of Equation (7), including independent and dependent coefficients, were calculated and checked by the Student criterion [30].

After calculating, a variance $s^2$ of 72.3 was calculated, the average t was determined to be 517.29, and the standard deviation s was computed as 8.5.

Based on the experimental data in Tables 2 and 3, the mathematical equation of forming pressure is shown in Equation (12).

$$Y = 510.22 - 16.8 \, x_1^2 + 22.8 \, x_2^2 + 84.3 \, x_1 + 25.58 \, x_2 + 31.58 \, x_3 + 13.75 \, x_1 x_2 + 23.75 \, x_1 x_3 - 20.5 x_2 x_3 \tag{12}$$

The suitability of Equation (15), determined by the Fisher criterion, is represented by Equation (13). It shows that the function is suitable.

$$F_\alpha = 5.56 \leq F_t = 19.33 \tag{13}$$

where $F_t$ is tabulated according to the Fisher criterion and $F_\alpha$ is the adequacy according to the Fisher criterion [30].

In order to determine the influence of each input parameter on the maximum forming pressure value, Equations (14)–(16) are rewritten from Equation (12) as follows:

$$Y_1(x_1, 1, 1) = 569.68 + 121.8x_1 - 16.8\, x_1{}^2 \tag{14}$$

$$Y_2(1, x_2, 1) = 633.05 + 18.83\, x_2 + 22.8\, x_2{}^2 \tag{15}$$

$$Y_3(1, 1, x_3) = 639.85 + 34.81\, x_3 \tag{16}$$

Based on the coefficients of Equations (14)–(16), it can be seen that the binder pressure $(x_1)$ is the parameter that has the greatest impact on the forming pressure. Next, the second most influential parameter is the workpiece thickness. The ratio of stress is the parameter that has the least influence on the value of the maximum forming pressure. A comparison of the value of the experiment with the value obtained from Equation (12) is shown in Figure 9. The error of the mathematical model and the experimental result is 2.4%. It can be seen that the difference between the experimental results and results calculated from the mathematical models is not significant.

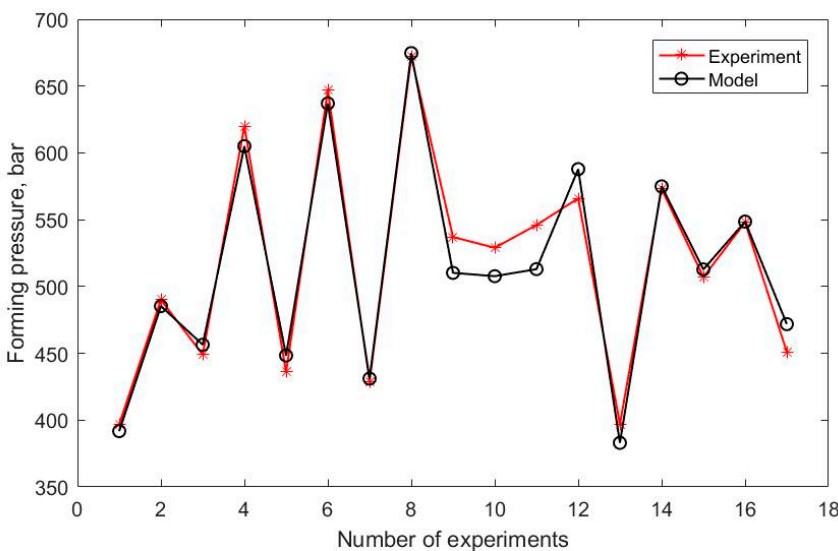

**Figure 9.** Comparison of modeled and experimental results.

The analysis results from the MATLAB software for Equation (12) show the influence of each factor on the forming pressure. The dependence of the maximum forming pressure on the thickness and the ratio of stress is shown in Figure 10, while the dependence of the maximum forming pressure on the binder pressure and the ratio of stress is shown in Figure 11; finally, the dependence of the maximum forming pressure on the binder pressure and the thickness is shown in Figure 12.

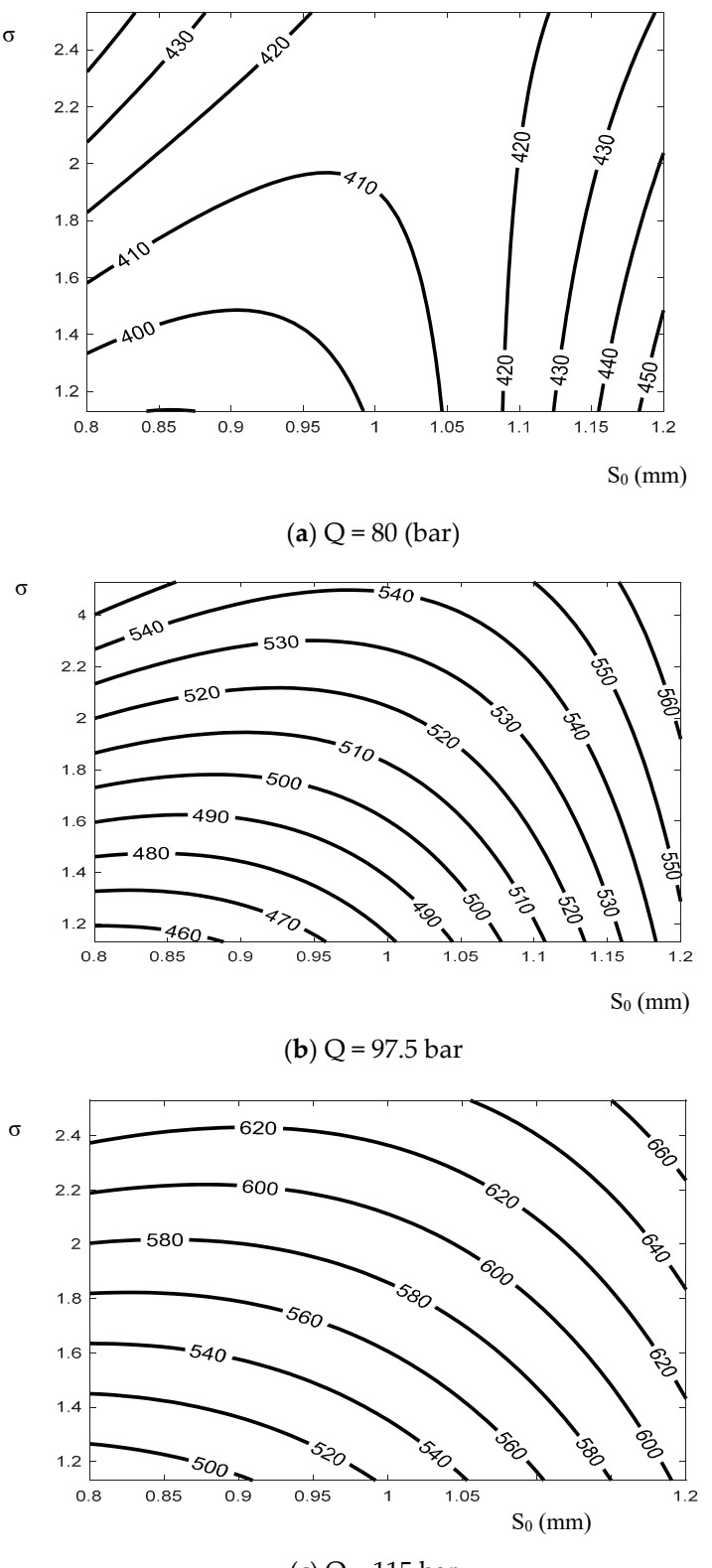

(**a**) Q = 80 (bar)

(**b**) Q = 97.5 bar

(**c**) Q = 115 bar

**Figure 10.** Dependence of the maximum forming pressure on the thickness and the ratio of stress.

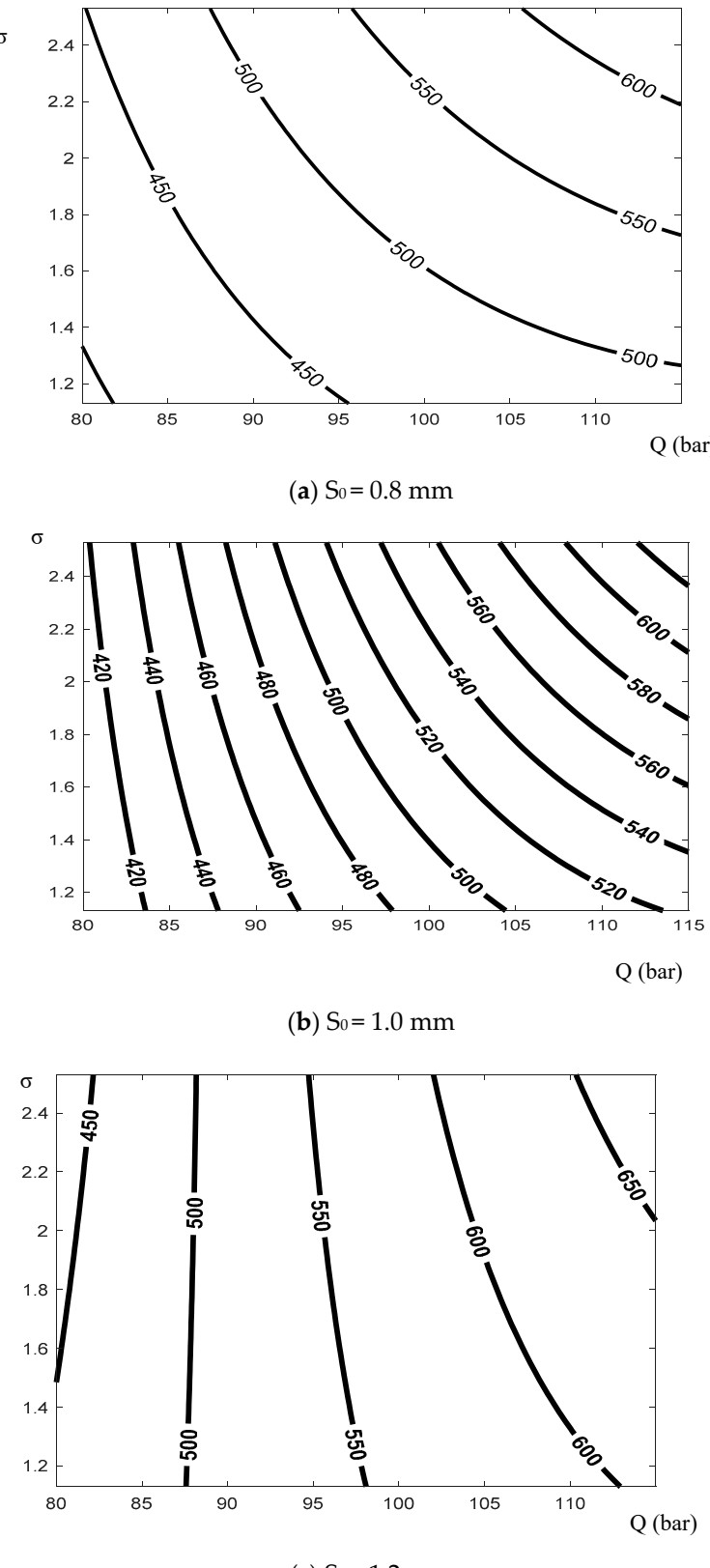

(**a**) $S_0 = 0.8$ mm

(**b**) $S_0 = 1.0$ mm

(**c**) $S_0 = 1.2$ mm

**Figure 11.** Dependence of the maximum forming pressure on the binder pressure and the ratio of stress.

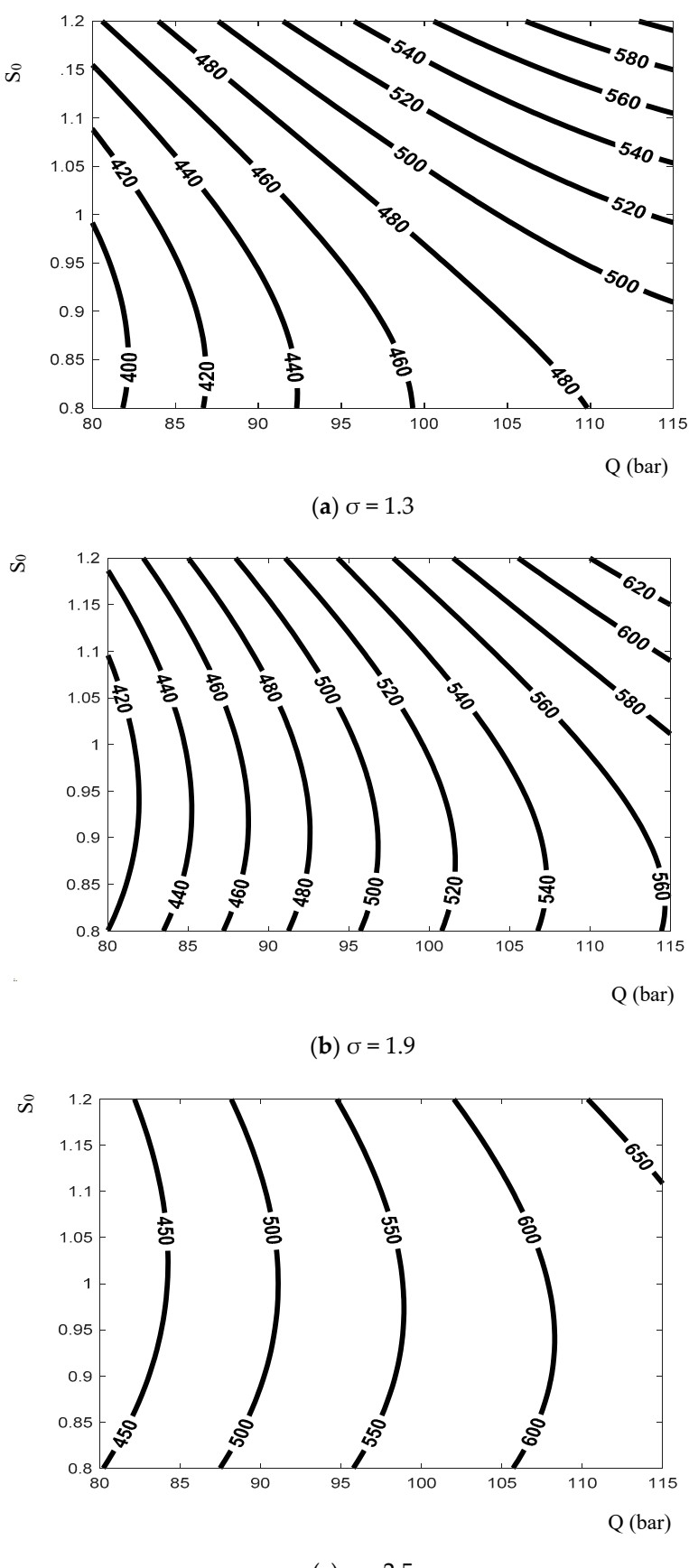

**Figure 12.** Dependence of the maximum forming pressure on the binder pressure and the thickness.

The graphs in Figures 10–12 show the dependence of forming pressure on each input parameter. The ratio of stress σ presents the deformation ability of the material. The larger the ratio, the more flexible the material, which means that this material is easy to deform. The graphs in Figures 10 and 11 show that the larger the ratio, the higher the maximum pressure. When materials are easily deformed, the ease with which they can be filled at the bottom radius of the die is higher. Therefore, the liquid pressure can be increased for formation without destroying the product.

The graphs in Figures 10 and 12 show the heterogeneous dependence of the fluid pressure on the initial workpiece thickness. Most diagrams indicate that there is a value of the workpiece thickness at which the obtained fluid pressure is minimal.

The graphs in Figures 11 and 12 illustrate that when the binder pressure increases, the forming liquid pressure also increases according to a certain rule. The cause of this phenomenon is that the workpiece is more strongly hampered in the forming process. Therefore, in order to pull the workpiece within the capacity of the die, the forming pressure needs to be increased appropriately.

In general, it is possible to evaluate the impact level of each factor (the binder pressure, the thickness of the workpiece, and the ratio of stress) on the forming liquid pressure parameters by using Equation (12). The impact rule of each factor is clarified, and therefore, it is possible to adjust the parameters to make the shaping process more stable.

## 5. Conclusions

In this paper, hydrostatic formation for cylindrical products with different kinds of material was investigated by experiment, where the relationship between the maximum forming pressure and some significant input parameters was clarified. The study showed the following novel results:

- A mathematical model has been constructed from experimental results based on the orthogonal second-order design method. Mathematical verifications have been carried out to confirm the high reliability of this model. The model shows the dependence of the maximum forming pressure on parameters including the binder pressure, the thickness of the workpiece, and the ratio of stress of each material.
- The research has shown that, among the three kinds of material considered, under the same defined conditions, the lowest critical pressure is found with copper formation, while the highest value corresponds to stainless steel.
- Among the investigated factors, the binder pressure is proven to be the factor that has the greatest impact on the magnitude of the forming pressure. The greater the binder pressure, the greater the ability to maintain the magnitude of the liquid pressure.
- The comparison between the mathematical model of the forming pressure and the experimental results is conducted, and a good agreement is demonstrated. Therefore, the model is meaningful for predicting the critical pressure and stabilizing the hydrostatic stamping forming process for the considered materials.

The research results have only been considered within defined experimental conditions. Therefore, to fully control the hydrostatic forming process, it is necessary to conduct further study into other factors such as friction, strain rate, and material thinning distribution.

**Author Contributions:** T.-K.L.: analysis of experimental data, and design of the experimental system. T.-T.N.: design and manufacture of a complete test system, and writing the original manuscript. N.-T.B.: writing—review and editing, and funding acquisition. All authors have read and agreed to the published version of the manuscript.

**Funding:** This research was funded by Hanoi University of Science and Technology (HUST) under project number T2020-TT-202.

**Institutional Review Board Statement:** Not applicable.

**Informed Consent Statement:** Not applicable.

**Acknowledgments:** This research was funded by Hanoi University of Science and Technology (HUST) under project number T2020-TT-202. This work was supported by the Centennial SIT Action for the 100th anniversary of Shibaura Institute of Technology entering the top 10 at the Asian Institute of Technology.

**Conflicts of Interest:** On behalf of all authors, the corresponding author states that there is no conflict of interest.

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
