# Peer review of "Experimental Modeling of Pressure in the Hydrostatic Formation of a Cylindrical Cup with Different Materials"

_applsci, doi:10.3390/app11135814_

Round 1
Reviewer 1 Report
See the attached file.

Author Response
"Please see the attachment."

Reviewer 2 Report
The article "Experimental modelling of forming pressure in hydrostatic forming for cylindrical cup with different materials" deals with a very interesting topic. However, the way it was prepared has resulted in research being very narrowly focused and not providing the necessary comprehensive data set.
- First of all, English must be improved, there are typos, missing articles, wrong order of words in sentences, etc.
- Figure 2, 8: missing scale
- Figure 7: what was the difference between forming pressure for the two samples?
- Figure 9: the image is of poor quality. Why is only 15 test pieces shown when 17 were planned?
- Line 243: You say that the most important parameter that affects the forming pressure is binder pressure and the second most important is the ratio of stress. How would this be compared to other parameters that were not considered in this study? For example, the radius?
- It would be appropriate to include more experimental results.
- The conclusion must be improved. It looks more like a general summary. It would be useful to add some specific values.
Author Response
"Please see the attachment."

Round 2
Reviewer 2 Report
Dear authors,
thank you for your revision of the paper.